**Original research**

# Health-seeking behaviours in a malaria endemic district in Lao People's Democratic Republic: a mixed methods study

Ken Ing Cherng Ong [ORCID],[1] Phonepadith Khattignavong,[2] Sengdeuane Keomalaphet,[2] Moritoshi Iwagami,[2,3] Paul Brey,[2] Shigeyuki Kano,[2,3] Masamine Jimba [ORCID][1]

¹Department of Community and Global Health, Graduate School of Medicine, The University of Tokyo, Tokyo, Japan
²Institut Pasteur du Laos, Vientiane, Lao People's Democratic Republic
³Department of Tropical Medicine and Malaria, National Center for Global Health and Medicine, Research Institute, Tokyo, Japan

**Correspondence to**
Ken Ing Cherng Ong;
kenicong@m.u-tokyo.ac.jp

## ABSTRACT

**Objectives** This mixed methods study was conducted to explore the barriers and facilitators for health-seeking behaviours in a malaria endemic district in Lao PDR.

**Design** A convergent mixed methods design.

**Setting** Two malaria endemic villages in Thapangthong district, Savannakhet Province, Lao PDR.

**Participants** Villagers and healthcare workers in the two villages in Thapangthong district.

**Methods** In the quantitative part, a pretested questionnaire was used to identify the health-seeking behaviours of the villagers. In the qualitative part, focus group discussions were employed to explore health-seeking behaviours of the villagers and in-depth interviews were used to explore the perceptions of the healthcare workers. Descriptive statistics were computed and multiple logistic regressions were used to identify the factors associated with perceived severity and perceived susceptibility. Thematic analysis was used to analyse the qualitative data. Quantitative and qualitative results were integrated in joint displays.

**Results** In the quantitative part, data were collected from 313 villagers from both villages. For malaria, 96.0% and 98.2% of villagers from villages A and B, respectively, would first seek treatment at public health facilities. Villagers who have not experienced malaria before were more likely to perceive that the consequences of malaria were serious compared with those who have experienced malaria before (adjusted OR=1.69, 95% CI: 1.03 to 2.75). However, qualitative data showed that villagers faced problems such as lack of medicines and medical equipment. Healthcare workers also mentioned the lack of manpower and equipment in the in-depth interviews. Nevertheless, villagers still preferred to seek treatment at the health center as the National Health Insurance was introduced.

**Conclusions** Public health facility usage was high but barriers existed. Effective policy and enabling environment such as the introduction of the National Health Insurance could help accelerate the progress towards the malaria elimination goal. Moreover, the benefits could go beyond the context of malaria.

## INTRODUCTION

Malaria threatens lives.[1] In 2018, 405 000 lives were lost to malaria and 228 million cases

---

### Strengths and limitations of this study

► This study used both quantitative and qualitative methods to explore the health-seeking behaviours in a malaria endemic district in Lao People's Democratic Republic (PDR).

► Perspectives from both the villagers and the healthcare providers were obtained and analysed.

► Baseline health-seeking behaviours data were not available but data integration indicated the improvement in health-seeking behaviours in the malaria endemic district.

► Generalisability might be an issue but the lessons learnt might be transferable to other malaria endemic areas in Lao PDR and other similar settings.

---

were reported.[2] Moreover, 272 000 deaths (67%) occurred in children under 5 years of age.[2] Currently, the effort to rid the world of malaria has regained momentum and an ambitious goal to eradicate malaria by 2050 has been set once again.[3] However, the application of the same, presently available tools and approaches will not help us achieve the goal.[4]

In the Greater Mekong Subregion (GMS), malaria mortality has substantially decreased in the past decade.[5] For example, from October to December 2018, the region reported 25 974 malaria cases, equivalent to a 21% decrease compared with the same period in the previous year.[6] However, malaria elimination efforts in many GMS countries have been hampered by difficulties, such as large movements of people across borders, diverse geographical terrains and the availability and access to counterfeit antimalarials.[7] Furthermore, these factors have also contributed to the emergence of antimalarial drug resistance.[8]

In the Lao People's Democratic Republic (Lao PDR), which is situated in the GMS, the key risk populations have been identified as ethnic minority groups living in remote forested and mountainous areas, plantation workers, migrant workers, and the soldiers.[9][10] Lao PDR is a linguistically and ethnically diverse nation.[11] Ethnic minorities in Lao PDR usually inhabit remote mountainous and forest areas and are usually characterised by extreme poverty.[9] Access to routine healthcare services is also limited for these ethnic minorities due to geographical barriers.[7] These ethnic minorities usually have their own distinct languages and many do not speak Lao.[5][9] As a result, dissemination of health messages is challenging.[9] Moreover, many ethnic minorities are animists with their own unique beliefs towards illness such as evil spirits could cause malaria.[11][12] Consequently, many people from these groups still resort to traditional medicine and religious/animistic rituals for treatment.[12]

The level of knowledge on modern medicine in Lao PDR is still low.[13][14] To complicate matters further, substandard and counterfeit medicines are still available in private pharmacies.[13][15] In the context of malaria, this could hamper the ongoing efforts of malaria elimination, endanger patients' lives, and worsen the drug resistance problem in Lao PDR.[13][15]

To achieve the Sustainable Development Goal 3: Good Health and Well-being for All; a key element must be attained: achieving Universal Health Coverage (UHC). To this end, the Lao Government has introduced the National Health Insurance (known by the locals as *Kor Por Sor*) scheme in 2016. In this scheme, all Lao citizens can receive treatments at public health facilities at a very minimal cost. For example, for outpatient visits at a village health center, villagers pay only 5000 kip (approximately US$ 0.63, as of January 2019) per visit. However, the implementation of the National Health Insurance scheme still faces barriers such as low awareness of this scheme among the villagers.[16]

To accelerate the progress towards malaria elimination by 2030, understanding the level of knowledge of the populations in malaria endemic areas on modern medicine and their health-seeking behaviours is critical to inform the policymakers on the types of interventions and treatment plans and policies to improve healthcare delivery in Lao PDR. The objective of this study was to explore health-seeking behaviours from the perspectives of the villagers and the frontline healthcare workers in a malaria endemic area in Lao PDR.

## METHODS
### Study design and settings
A convergent mixed methods design was used where both quantitative and qualitative data were simultaneously collected and integrated.[17] In this study, quantitative data were collected from the villagers, and qualitative data from both the villagers and the healthcare workers.

This study was conducted in October 2018 in Thapangthong district, Savannakhet province, Lao PDR. This district was chosen because of the high number of reported malaria cases. In 2018, out of 2245 reported malaria positive cases in Savannakhet province, 323 were from Thapangthong district (Data obtained by personal communication with the Center of Malariology, Parasitology and Entomology, Lao PDR). Two villages (villages A and B) were chosen after consultation with the District Health Office in Thapangthong district based on the number of highest malaria cases and consultation with the village headmen.

Thapangthong district comprises 42 villages, and the population as of October 2018 was 42 563, of which 21 157 were female villagers. The district has one district hospital and the ethnic composition of this district is 42% Lao ethnic group and 58% Katang ethnic group. The population of village A was 1149 villagers living in 215 households. Out of 1149 villagers, 570 were women. The major ethnic group in village A is Lao. Village B has 753 villagers living in 125 households. Out of 753 villagers, 324 were women. The major ethnic group in village B is Katang.

### Study participants and recruitment
Study participants in this study were villagers and healthcare workers aged 18 years and above living and working in the two villages. After village A and village B were chosen, the village headmen of each village were informed of the purpose of this study. For quantitative data collection, the village headmen informed the whole village a few days before the study took place, and requested a representative of each household who agreed and were willing to participate in the study to come at the specified time and place (either at the village meeting room or the meeting room of the health center).

For qualitative data collection, villagers who finished the quantitative interview were personally approached and informed about the focus group discussions (FGDs). All the villagers who volunteered were included. For the in-depth interviews (IDIs) of the healthcare workers, all the healthcare workers were approached and informed of the purpose of the study. All those who agreed to participate were interviewed.

### Quantitative part
A structured questionnaire was created by adapting the content from the latest Lao Social Indicator Survey,[18] Malaria Indicator Survey from Roll Back Malaria[19][20] and clinical vignettes from a study by Mebratie *et al*.[21] The questionnaire consisted of seven sections: health-seeking behaviours, knowledge on malaria and modern medicine, perceived severity and perceived susceptibility, perceived benefits, self-efficacy, clinical vignettes, and socioeconomic characteristics.

The questionnaires were translated into Lao by two authors (PK and SKe, both medical doctors and malaria researchers) who are Lao native speakers and fluent in English. They also checked each other's translation for

accuracy and checked the questionnaire for content and face validity. Before data collection, six research assistants, who were medical doctors or healthcare worker, were trained on the content of the questionnaire and ethical issues such as privacy protection of the participants. The questionnaire was pretested in another village in the same district among 14 villagers. After the pretest, a meeting with all the research assistants was held to revise the questionnaires for easier understanding based on their feedbacks.

Six research assistants collected the quantitative data by interviewer-administered questionnaire using KoBo Collect application downloaded on password protected Android tablets provided to each research assistant. Each villager who finished answering the questionnaire was provided with a bag of sanitary items such as soap and washing detergents as an incentive.

Descriptive statistics were computed to summarise the sociodemographic and socioeconomic characteristics of the participants. $\chi^2$ test was used for categorical variables. Multiple logistic regressions were used to identify the factors associated with perceived severity and perceived susceptibility. In both models, the explanatory variables were village, sex, ethnicity, malaria experience and knowing that malaria is caused by mosquito bites. For clinical vignettes, providers such as village health volunteers, health centers, district hospitals, and provincial hospitals were categorised as 'public health facilities'. Other providers were grouped as 'others'. Statistical significance was set at p<0.05. All statistical analyses were performed with Stata/IC V.13.1 (College Station, Texas, USA). This study is in accordance with the Strengthening the Reporting of Observational studies in Epidemiology Statement checklist (online supplemental file 1).[22]

### Qualitative part

FGDs were conducted among villagers and IDIs among healthcare workers. Three authors (KICO, PK, and SKe) collected data until saturation was reached, meaning no new information emerging from the study participants even after additional FGDs or IDIs were conducted.

FGD and IDI topic guides were created based on the quantitative questionnaire for the villagers. The FGD topic guide prompted the villagers to elaborate their experience with malaria, their choice of treatment, the reasons behind their decisions, health-seeking behaviours in general and the cause of malaria. As for the healthcare workers, the IDI topic guide asked about the health-seeking behaviours of the villagers, and the barriers in providing proper care.

All FGDs and IDIs were audio recorded and field notes were taken. Each villager and healthcare worker who finished the FGD or IDI was provided with a bag of sanitary items such as soap and washing detergents as an incentive. Audio data were transcribed verbatim in Lao language by native Lao research assistants. All the transcripts were double checked for accuracy. Thematic analysis was used to analyse the data and was performed by three authors (KICO, PK, and SKe). To ensure trustworthiness in the analysis, the steps outlined by Nowell et al[23] were followed. NVivo V.12 Pro (QRS International Pty, Doncaster, Australia) software was used to expedite the data analysis process. Guidelines in the Consolidated Criteria for Reporting Qualitative Research were followed (online supplemental file 2).[24]

### Data integration

Both the quantitative data and qualitative data were integrated to gain new insights through joint displays. A joint display is a visual means to combine both qualitative and quantitative data.[25 26]

### Ethical considerations

Participation in this study was voluntary and the research assistants explained the purpose of the study prior to the data collection to the villagers and healthcare workers. A written informed consent was obtained from each participant. Personally identifiable information was not collected and the IDIs and FGDs were conducted in a secured place with low risk of being overheard by outsiders, such as the village meeting room and the meeting room of the health center.

## RESULTS
### Quantitative part

Table 1 shows the sociodemographic characteristics of the villagers. In total, data were collected from 313 villagers; 202 from village A and 111 from village B. The number of female villagers in village A was 128 (63.4%), while in village B, it was 44 (39.6%). In village A, 198 (98.0%) villagers were Buddhists, while in village B, 81 (73.0%) were animists. The major ethnic group in village A was Lao (197, 97.5%) while that of village B was Katang (89, 80.2%). In both villages, the majority of the villagers were farmers (village A: 170 (84.2%); village B: 105 (94.6%)).

Table 2 shows the first place to seek treatment when a family member falls sick in both villages. Half of the villagers in village A (101, 50.0%) would go to the health center first compared with 105 (94.6%) villagers in village B.

Table 3 shows the results of knowledge of malaria and treatment among the villagers. In village A, 115 (56.9%) villagers answered that malaria is caused by mosquito bites compared with 85 (76.6%) villagers in village B. Drinking dirty water was mentioned as a cause of malaria by 30 (14.9%) villagers in village A and 27 (24.3%) villagers in village B. In both villages, no villagers attributed witchcraft to be the cause of malaria. Regarding protection against malaria, 150 (74.3%) villagers in village A answered sleeping under an insecticide-treated bed net compared with 101 (91.0%) villagers in village B. In village A, 23 (11.4%) villagers thought that not drinking dirty water could protect oneself from malaria compared with 28 (25.2%) villagers in village B. Most villagers in

**Table 1**  Sociodemographic characteristics of the villagers

|  |  | Village A | Village B |
|---|---|---|---|
| Villagers interviewed |  | 202 | 111 |
| Sex, n (%) |  |  |  |
|  | Male | 74 (36.6) | 67 (60.4) |
|  | Female | 128 (63.4) | 44 (39.6) |
| Age |  |  |  |
|  | Minimum | 18 | 18 |
|  | Maximum | 85 | 73 |
|  | Median | 40 | 39 |
|  | Mean | 40.9 | 38.6 |
| Religion, n (%) |  |  |  |
|  | Buddhism | 198 (98.0) | 30 (27.0) |
|  | Animism | 4 (2.0) | 81 (73.0) |
| Ethnicity, n (%) |  |  |  |
|  | Lao | 197 (97.5) | 21 (18.9) |
|  | Katang | 2 (1.0) | 89 (80.2) |
|  | Do not know | 3 (1.5) | 1 (0.9) |
| Marital status, n (%) |  |  |  |
|  | Married | 183 (90.6) | 97 (87.4) |
|  | Single | 5 (2.5) | 6 (5.4) |
|  | Divorced | 2 (1.0) | 1 (0.9) |
|  | Widowed | 12 (5.9) | 6 (5.4) |
|  | Others | 0 | 1 (0.9) |
| Occupation, n (%) |  |  |  |
|  | Farmer | 170 (84.2) | 105 (94.6) |
|  | Soldier | 1 (0.5) | 0 |
|  | Housewife | 18 (8.9) | 2 (1.8) |
|  | Businessman/ women | 1 (0.5) | 1 (0.9) |
|  | Government servant | 7 (3.5) | 2 (1.8) |
|  | Others | 5 (2.5) | 1 (0.9) |
| Education, n (%) |  |  |  |
|  | No education | 71 (35.2) | 42 (37.8) |
|  | Preschool | 1 (0.5) | 1 (0.9) |
|  | Primary school | 84 (41.6) | 46 (41.4) |
|  | Lower secondary school | 28 (13.9) | 14 (12.6) |
|  | Upper secondary school | 8 (4.0) | 5 (4.5) |
|  | Post secondary non-tertiary | 8 (4.0) | 2 (1.8) |
|  | Tertiary | 2 (1.0) | 1 (0.9) |
| Average monthly household income (kip) (US\$ 1=8000 kip (January 2019)) |  |  |  |
|  | Minimum | 0 | 0 |
|  | Maximum | 10 000 000 | 4 000 000 |
|  | Median | 200 000 | 100 000 |
|  | Mean | 496 881 | 290 000 |
| Number of household members (median) |  | 5 | 6 |

Continued

**Table 1**  Continued

| Have insecticide-treated nets at home |  |  |  |
|---|---|---|---|
|  | Yes | 169 (83.7) | 107 (96.4) |
|  | No | 31 (15.4) | 4 (3.6) |
|  | Do not know | 2 (1.0) | 0 |

*Percentage may not add up to 100 due to rounding.

both villages did not know the most effective treatment for malaria (village A: 185 (91.6%); village B: 85 (76.6%)).

Table 4 shows the individual beliefs of the villagers in the context of malaria. In village A, 109 (54.0%) villagers perceived that the consequences of malaria were serious compared with 60 (54.1%) villagers in village B. Regarding perceived susceptibility, 112 (55.5%) villagers in village A perceived that they were at high risk of malaria compared with 61 (55.0%) in village B. As for perceived benefits, bed net use was thought to reduce malaria risk by 126 (62.4%) villagers in village A compared with 71 (64.0%) villagers in village B.

Table 5 shows the responses of the villagers to clinical vignettes describing five conditions: haemolytic anaemia, malaria, tuberculosis, acute respiratory infection and diarrhoea. For malaria, 194 (96.0%) villagers in village A responded that they would first seek treatment at public health facilities, while 109 (98.2%) villagers in village B responded that they would do the same. In village B, 97%–98% of the villagers answered that they would first seek treatment at the public health facilities for other conditions.

Table 6 shows the factors associated with both perceived severity of malaria and perceived susceptibility of malaria. For perceived severity of malaria, those who have not experienced malaria before were more likely to perceive that the consequences of malaria were serious compared with those who have experienced malaria before (adjusted OR=1.69, 95% CI: 1.03 to 2.75). Village, sex, ethnicity, malaria experience and knowing about the cause of malaria were not significantly associated with perceived susceptibility.

**Table 2**  First place to seek treatment when a family member falls sick

| Where to seek treatment, n (%) | Village A (n=202) | Village B (n=111) |
|---|---|---|
| Village health volunteer | 25 (12.4) | 0 |
| Health center | 101 (50.0) | 105 (94.6) |
| District hospital | 46 (22.8) | 4 (3.6) |
| Provincial hospital | 6 (3.0) | 1 (0.9) |
| Private pharmacy | 10 (5.0) | 0 |
| Traditional healer | 1 (0.5) | 0 |
| Others | 13 (6.4) | 1 (0.9) |

**Table 3** Knowledge of malaria and treatment

| | Village A (n=202) | Village B (n=111) |
|---|---|---|
| **What causes malaria? n (%)** | | |
| Mosquito bites | 115 (56.9) | 85 (76.6) |
| Eating dirty food | 17 (8.4) | 9 (8.1) |
| Eating uncooked food | 6 (3.0) | 7 (6.3) |
| Drinking dirty water | 30 (14.9) | 27 (24.3) |
| Witchcraft | 0 | 0 |
| Going to the forest | 73 (36.1) | 49 (44.1) |
| Changing weather | 8 (4.0) | 3 (2.7) |
| Getting soaked with rain | 3 (1.5) | 1 (0.9) |
| **How to protect yourself against malaria? n (%)** | | |
| Sleep under an insecticide-treated mosquito net | 150 (74.3) | 101 (91.0) |
| Use mosquito repellent | 9 (4.5) | 4 (3.6) |
| Avoid mosquito bites | 9 (4.5) | 5 (4.5) |
| Avoid going to the forest | 10 (5.0) | 4 (3.6) |
| Take preventive medicine | 2 (1.0) | 0 |
| Spray house with insecticide | 8 (4.0) | 1 (0.9) |
| Use mosquito coils | 10 (5.0) | 4 (3.6) |
| Fill in puddles around the house | 18 (8.9) | 3 (2.7) |
| Keep surrounding of the house clean | 54 (26.7) | 42 (37.8) |
| Burn leaves | 4 (2.0) | 10 (9.0) |
| Do not drink dirty water | 23 (11.4) | 28 (25.2) |
| Do not eat dirty food | 8 (4.0) | 12 (10.8) |
| Put mosquito screens on the window | 0 | 1 (0.9) |
| Do not get soaked with rain water | 0 | 0 |
| **What are the symptoms that indicate a person has malaria? n (%)** | | |
| Fever | 115 (56.9) | 70 (63.1) |
| Feeling cold | 89 (44.1) | 73 (65.8) |
| Headache | 116 (57.4) | 85 (76.6) |
| Nausea and vomiting | 30 (14.9) | 17 (15.3) |
| Diarrhoea | 2 (1.0) | 2 (1.8) |
| Dizziness | 16 (7.9) | 13 (11.7) |
| Loss of appetite | 12 (5.9) | 4 (3.6) |
| Bodyache/joint pain | 58 (28.7) | 23 (20.7) |

Continued

**Table 3** Continued

| | Village A (n=202) | Village B (n=111) |
|---|---|---|
| Pale eyes | 5 (2.5) | 0 |
| Feeling weak | 14 (6.9) | 9 (8.1) |
| **What is the most effective medication to treat malaria? n (%)** | | |
| Chloroquine | 0 | 1 (0.9) |
| Quinine | 2 (1.0) | 3 (2.7) |
| Artemisinin combination therapy (Coartem TM) | 3 (1.5) | 12 (10.8) |
| Aspirin, panadol and paracetamol | 12 (5.9) | 8 (7.2) |
| Do not know | 185 (91.6) | 85 (76.6) |
| **What are the main danger signs of malaria? n (%)** | | |
| Seizure | 29 (14.4) | 16 (14.4) |
| Fainting | 7 (3.5) | 5 (4.5) |
| Any fever | 8 (4.0) | 1 (0.9) |
| High fever | 26 (12.9) | 20 (18.0) |
| Stiff neck | 2 (1.0) | 3 (2.7) |
| Feeling weak | 8 (4.0) | 3 (2.7) |
| Not active | 2 (1.0) | 2 (1.8) |
| Chills/shivering | 30 (14.9) | 24 (21.6) |
| Not able to eat | 4 (2.0) | 1 (0.9) |
| Vomiting | 9 (4.5) | 18 (16.2) |
| Crying all the time | 4 (2.0) | 1 (0.9) |
| Restless | 19 (9.4) | 14 (12.6) |
| Diarrhoea | 0 | 2 (1.8) |

## Qualitative part

In village A, two male and two female FGDs were conducted, while in village B, three FGDs each for male and female villagers were conducted. The number of villagers per group was 3–4. Online supplemental files 3, 4 show the characteristics of the villagers who participated in the FGDs and healthcare workers who participated in the IDIs. The FGDs and IDIs lasted about an hour. Themes that emerged from the FGDs among the villagers are shown in (online supplemental file 5).

### Theme: attributing the cause of malaria to something else other than being bitten by mosquitoes

Some villagers believed that malaria was caused by something else other than mosquito bites. For example, some villagers mentioned ghosts or unclean water as the cause of malaria.

> When I had malaria, they used 'pee pop' (ghost) to exorcize it away. (village A, female, 30s, farmer)

**Table 4** Individual beliefs

| | | Village A (n=202) | Village B (n=111) | P value |
|---|---|---|---|---|
| Villagers who perceived the consequences of malaria were serious (perceived severity) n (%) | | 109 (54.0) | 60 (54.1) | 0.987 |
| Villagers who perceived they were at high risk of malaria (perceived susceptibility) n (%) | | 112 (55.5) | 61 (55.0) | 0.933 |
| Villagers who believed that the recommended practice or product will reduce their risk (perceived benefits) n (%) | | | | |
| | Bed net use | 126 (62.4) | 71 (64.0) | 0.781 |
| | Malaria diagnostics | 120 (59.4) | 64 (57.7) | 0.764 |
| | Malaria treatments with ACT | 159 (78.7) | 88 (79.3) | 0.906 |
| Villagers who are confident in their abilities to perform a specific malaria-related behaviour (self-efficacy) n (%) | | | | |
| | Protection of self and family | 158 (78.2) | 76 (68.5) | 0.057 |
| | Bed net use | 198 (98.0) | 110 (99.1) | 0.466 |
| | Malaria detection | 153 (75.7) | 81 (73.0) | 0.589 |
| | Seek diagnosis | 185 (91.6) | 96 (86.5) | 0.154 |
| | Seek treatment | 198 (98.0) | 107 (96.4) | 0.384 |

$\chi^2$ test.
ACT, artemisinin combination therapy.

Drinking unclean water, you will get malaria. (village B, male, 10s, farmer)

### Theme: attributing the cause of malaria to being bitten by mosquitoes

In contrast with the previous theme, some villagers correctly attributed the cause of malaria to being bitten by mosquitoes.

Because of mosquito bites. When you do not sleep under the bed net, the mosquitoes will bite you. (village B, male, 20s, government servant)

### Theme: describing an illness using local terms

The villagers also described common illnesses in the area using local terms and local languages.

'Kai luad niao' (sticky blood fever), when you are dehydrated, your blood will be sticky. When you drink water, your blood will dilute. (village A, female, 40s, farmer)

'Kai nyoong' (literally translated from Lao as 'mosquito fever', usually referred to malaria) in Katang language is called 'ae muay'. (village B, female, 40s, farmer)

### Theme: expressing hope to improve the health center

In the FGDs, the villagers expressed their hopes that the health center would be improved.

I hope the staff will be more knowledgeable on the treatments. (village B, male, 30s, farmer)

I hope the village health cente will be cleaner. There are mosquitoes here at the health center too, so you might even get malaria here! (village B, female, 20s, farmer)

### Theme: expressing difficulties in getting health services

The villagers also mentioned about their difficulties in getting health services. Some of the difficulties mentioned were financial or road conditions. In the following quote, the male villager used a local saying to illustrate that lack of financial resources could hinder one from getting medical services.

'Kon luai bor dai kau kuk, kon tuk bor dai kau hong mor'. (The rich do not go to jail, the poor do not go to the hospital) (village B, male, 20s, farmer)

I hope the roads will also be improved. When it is raining, people die because they are not sent to the hospital in time. (village B, male, 30s, farmer)

### Theme: recalling personal experience

Although the district has one district hospital and village health centers, these facilities were not equipped to handle complicated cases and the patients must be referred to the provincial hospital, which is a 5-hour to 7-hour drive in the dry season and longer in the rainy season.

**Table 5** Clinical vignettes: first place to seek treatment

| | Haemolytic anaemia (n (%)) | | Malaria (n (%)) | | Tuberculosis (n (%)) | | Acute respiratory infection (n (%)) | | Diarrhoea (n (%)) | |
|---|---|---|---|---|---|---|---|---|---|---|
| | Village A (n=202) | Village B (n=111) | Village A (n=202) | Village B (n=111) | Village A (n=202) | Village B (n=111) | Village A (n=202) | Village B (n=111) | Village A (n=202) | Village B (n=111) |
| Public health facilities | 190 (94.1) | 109 (98.2) | 194 (96.0) | 109 (98.2) | 189 (93.6) | 108 (97.3) | 181 (89.6) | 109 (98.2) | 188 (93.1) | 108 (97.3) |
| Others | 9 (4.5) | 1 (0.9) | 7 (3.5) | 1 (0.9) | 11 (5.5) | 2 (1.8) | 20 (9.9) | 1 (0.9) | 12 (5.9) | 2 (1.8) |
| Take no action | 3 (1.5) | 1 (0.9) | 1 (0.5) | 1 (0.9) | 2 (1.0) | 1 (0.9) | 1 (0.5) | 1 (0.9) | 2 (1.0) | 1 (0.9) |

**Table 6** Factors associated with perceived severity and perceived susceptibility

| Perceived severity of malaria | | |
|---|---|---|
| **Characteristics** | **Adjusted OR (95% CI)** | **P value** |
| Village | | |
| Village A | 1.00 | 0.811 |
| Village B | 1.11 (0.48 to 2.58) | |
| Sex | | |
| Male | 1.00 | 0.090 |
| Female | 0.65 (0.40 to 1.07) | |
| Ethnicity | | |
| Lao | 1.00 | 0.615 |
| Others | 0.80 (0.34 to 1.89) | |
| Experienced malaria before | | |
| Yes | 1.00 | **0.036** |
| No | 1.69 (1.03 to 2.75) | |
| Knowing that malaria is caused by mosquito bites | | |
| Yes | 1.00 | 0.217 |
| No | 0.74 (0.46 to 1.20) | |
| **Perceived susceptibility of malaria** | | |
| **Characteristics** | **Adjusted OR (95% CI)** | **P value** |
| Village | | |
| Village A | 1.00 | 0.805 |
| Village B | 0.90 (0.39 to 2.07) | |
| Sex | | |
| Male | 1.00 | 0.910 |
| Female | 1.03 (0.63 to 1.67) | |
| Ethnicity | | |
| Lao | 1.00 | 0.754 |
| Others | 1.14 (0.49 to 2.67) | |
| Experienced malaria before | | |
| Yes | 1.00 | 0.344 |
| No | 1.26 (0.78 to 2.04) | |
| Knowing that malaria is caused by mosquito bites | | |
| Yes | 1.00 | 0.611 |
| No | 0.88 (0.55 to 1.43) | |

We do not have oxygen here in the district hospital. I gave birth to a premature baby once and it died because we did not have oxygen. We were going to the Savannakhet provincial hospital and the baby died on the way in Pakxong. No incubator too. (village A, female, 30s, businesswoman)

### Theme: mentioning about the National Health Insurance (*Kor Por Sor*)

However, the villagers mentioned that the recently introduced National Health Insurance (*Kor Por Sor*) was a facilitator in seeking treatment from the health center.

> I will go to the health cente first, because I have to pay only 5000 kip. If I go to a private clinic, I will have to pay at least 50 000 kip. (village A, female, 30s, farmer)

Three IDIs were conducted in village A (two for healthcare workers and one for village health volunteer) and four were conducted in village B (three for healthcare workers and one for village health volunteer). The major themes were 'Mentioning about the National Health Insurance (*Kor Por Sor*)' and 'Expressing hope to improve the health center'. Online supplemental file 6 shows all other themes that emerged during the IDIs with the healthcare workers.

### Theme: mentioning about the National Health Insurance (*Kor Por Sor*)

When asked why the villagers come to the health centers, the healthcare workers also mentioned the National Health Insurance (*Kor Por Sor*) as the facilitator.

> Now there is the 'Kor Por Sor' (National Health Insurance) system, where you have to pay only 5000 kip, they will come even when they have common cold. (village A, female, 30s)

> Because of 'Kor Por Sor' (National Health Insurance), more people come to seek treatment at this health center. (village B, male, 30s)

### Theme: expressing hope to improve the health center

The healthcare workers also mentioned about their hopes to improve the health center and also improve their skills and knowledge to better serve the people.

> The knowledge and capability. I want to learn more. Another thing is the building. Too small. When people are sick they can't sleep here. If possible, I hope for more advanced equipment too. (village B, male, 30s)

> At this health center the manpower is not enough, we also do not have enough knowledge. I hope people with knowledge will come here to work at this health center. I also hope this health center will be expanded. It is very crowded when there are many sick people and people who give birth. (village B, female, 20s)

### Data integration

Figures 1–3 show the joint displays combining both quantitative and qualitative data. Although in both villages, more than half of the villagers knew that malaria is caused by mosquito bites, some villagers still attributed the cause of malaria to drinking dirty water or water contaminated with mosquito larvae (figure 1).

When probed about the first place to seek treatment when a family member was sick, 72.8% from village A and 98.2% from village B indicated that they would go to either the health center or district hospital. In the context of malaria, a vast majority, 96.0% in village A and 98.2% in village B answered that they would first seek treatment at the public health facilities. In the qualitative part, both villagers and healthcare workers mentioned that after the introduction of the National Health Insurance (*Kor Por Sor*), the treatment cost became cheaper and this was the facilitator for the villagers to seek treatment at the health center or the district hospital (figure 2).

Although most of the villagers seek treatment from the health center or the district hospital, and are confident to seek treatment when having malaria (98.0% from village A and 96.4% from village B), many problems and dissatisfactions exist. For example, the lack of medication and proper equipment and the attitude of the healthcare workers might discourage the villagers from seeking proper healthcare treatment (figure 3).

## DISCUSSION

This study has three major findings. First, in the quantitative part, most of the villagers in both villages would seek treatment at the health center or the district hospital both in the context of malaria and other common diseases in the area. The level of knowledge of the villagers on the cause of malaria was low and the tendency to link malaria to causes other than mosquito bites was common. Second, in the qualitative part, both villagers and healthcare workers attributed increase in the number of villagers seeking treatment at the health center or the district hospital to the introduction of the National Health Insurance (*Kor Por Sor*). However, FGDs and IDIs revealed problems faced by the villagers in seeking treatment and healthcare workers in providing care. Third, even though the introduction of the National Health Insurance (*Kor Por Sor*) encouraged many villagers to seek treatment from the health center or the district hospital, both villagers and the healthcare workers were still facing hurdles in getting and providing optimal care. Although the two villages vary in terms of ethnic groups and religions, the health-seeking behaviours and the barriers they faced in receiving optimal healthcare services were similar.

In this study, villagers from both villages mainly sought treatment from the public healthcare providers: village health volunteer, health center or the district hospital. In Lao PDR, health-seeking behaviours are influenced by several factors such as distance to a health center, socioeconomic factors and family or friends' influence.[27 28] Compared with another study in Nong District, Savannakhet Province,[28] the number of villagers seeking treatment from traditional healers in this study was very low to none. The high usage of village public healthcare providers in this study might be due to the ease in accessing these public healthcare providers as the health centers of both villages were situated in the villages.[29] Moreover, in

| Quantitative Constructs | Qualitative Constructs |
|---|---|
| 56.9% in Village A and 76.6% in Village B attributed the cause of malaria to mosquito bites. | Theme: Attributing the cause of malaria to something else other than being bitten by mosquitoes. |
| 14.9% in Village A and 24.3% in Village B attributed the cause of malaria to drinking dirty water. | |
| *(From Table 3 Knowledge of malaria and treatment)* | *"Because your house is not clean. If you drink water with mosquito eggs, you will get fever."* Village A, female, 40s, government servant. |
| | *"Drinking unclean water, you will get malaria."* Village B, male, 10s, farmer. |

**Comments (meta-inferences)**

Many villagers still attributed the main cause of malaria to drinking dirty water or water contaminated with mosquito larvae. Despite efforts by so many stakeholders such as the WHO to eliminate malaria, misconception about the cause of malaria among the villagers can greatly hamper the progress in malaria elimination.

**Figure 1** Joint display of knowledge of the cause of malaria.

another study in Lao PDR, people who initially visited a public healthcare provider were more likely to maintain a connection with the provider.[30]

In the FGDs of the villagers and IDIs with the healthcare workers, one of the major reasons that increased the number of villagers seeking treatment at the health center or the district hospital was the introduction of the National Health Insurance (*Kor Por Sor*). In this scheme, villagers pay only 5000 kip (approximately US$ 0.63, as of January 2019) to get basic treatment and medication at the health center and 10 000 kip (approximately US$ 1.25, as of January 2019) at the district hospital.[16] Patients only need to bring the family registration book to prove their residence in the village. For those households without the family registration book, a special certificate issued by the village head is also acceptable (Personal communication with a Lao counterpart, based on an official Ministerial Order document). In Lao PDR, although malaria diagnosis and treatment are free, patients had to pay for treatment of other illnesses and those from the low-income bracket could not afford the treatment.[31] As economic status is a major barrier to getting proper treatment, the National Health Insurance has made it more affordable and accessible for the villagers to get treatment at the health center or the district hospital.

In this study, many villagers still attributed the cause of malaria to factors other than being bitten by mosquitoes, such as drinking unclean water. This result is consistent with another study among military personnel in Lao PDR.[31] Having knowledge on the cause of malaria does not necessarily lead to protective actions, but having knowledge can be a facilitator for self-implemented protective measures against malaria.[32] In the FGDs in both villages, many villagers described malaria and symptoms using local terms. This result emphasises the need to tailor health messages and elimination strategies that are locally acceptable and understandable.[12 33]

| Quantitative Constructs | Qualitative Constructs |
|---|---|
| 72.8% in Village A and 98.2% in Village B would first seek treatment at the health center or district hospital when someone in the family was sick.<br><br>*(From Table 2 First place to seek treatment when a family member falls sick)*<br><br>Responses to the malaria clinical vignette indicated that 96.0% in Village A and 98.2% in Village B would first seek treatment at the public health facilities.<br><br>*(From Table 5 Clinical vignettes)* | Theme: Mentioning about the National Health Insurance ("*Kor Por Sor*")<br><br>*"I will go to the health center first, because I have to pay only 5000 kip. If I go to a private clinic, I will have to pay at least 50000 kip."* Village A, female, 30s, farmer.<br><br>*"Because of 'Kor Por Sor' (National Health Insurance), we have more patients who come here for treatment."* Village B, female, 30s, healthcare worker. |
| **Comments (meta-inferences)**<br>The recent introduction of the National Health Insurance ("*Kor Por Sor*") seemed to have a positive impact as the financial burden on the villagers is greatly reduced. | |

**Figure 2** Joint display of the impact of National Health Insurance.

In addition, villagers who have not experienced malaria before in this study were more likely to perceive malaria to be more severe than those who experienced before. Asymptomatic malaria carriers are very common in endemic areas in Lao PDR and these carriers do not exhibit the usual malaria symptoms due to repeated infections.[34 35] Thapangthong district was no exception and those who had experienced malaria before could be asymptomatic carriers. This result also highlights the necessity to tackle asymptomatic infections in Lao PDR in order to achieve the elimination goal.

Although the quantitative results showed high usage of health centers and the district hospital, both the FGDs with the villagers and IDIs with the healthcare workers revealed many barriers in treatment seeking and treatment delivering in the district. In this study, unavailability of medications and the lack of proper medical equipment were some of the major grievances of the villagers. Moreover, the current reporting system for malaria cases

and antimalarial stocks at the village level to the district level in many malaria endemic areas in Lao PDR is still paper-based. This could cause a logistical delay in delivering the antimalarial stocks from Vientiane Capital to the endemic areas. In addition, some villagers also mentioned the unpleasant experience due to the attitude of the healthcare providers. Moreover, some of the healthcare providers acknowledged that they were not skilled enough in providing all treatments and hoped for better training. Lao PDR still faces many challenges in building a high-quality and equitable primary healthcare centers throughout the country.[27] The major challenges are difficult geographical terrains, inadequate training of healthcare staffs, lack of qualified health workers, low salaries and poor morale among the staff.[27 36] Therefore, although the current usage of the health center and the district hospital is high, it is crucial to strengthen the capacity of the frontline healthcare facilities to monitor and eliminate malaria and improve the reporting

| Quantitative Constructs | Qualitative Constructs |
|---|---|
| 72.8% in Village A and 98.2% in Village B would first seek treatment in the health center or district hospital when someone in the family was sick.<br><br>*(From Table 2 First place to seek treatment when a family member falls sick)*<br><br>91.6% from Village A and 86.5% from Village B indicated that they were confident to seek malaria diagnosis.<br><br>98.0% from Village A and 96.4% from Village B indicated that they were confident to seek malaria treatment.<br><br>*(From Table 4 Individual beliefs)* | Theme: Expressing opinion on the health center<br>*"Overall everything is good but only one thing is that medicine is always not enough."* Village A, male, 40s, teacher.<br><br>Theme: Recalling personal experience<br>*"We do not have oxygen here in the district hospital. I gave birth to a premature baby once and it died because we did not have oxygen. We were going to the Savannakhet provincial hospital and the baby died on the way in Pakxong. No incubator too."* Village A, female, 30s, businesswoman. |
| **Comments (meta-inferences)**<br>The ill-equipped facilities are a letdown as prescriptions are not always available and complicated cases will be referred to the provincial hospital which is a 5- to 7-hour drive away in the dry season, and longer in the rainy season. These factors can discourage the villagers from seeking proper care. ||

**Figure 3** Joint display of the barriers in treatment seeking.

mechanism by introducing online reporting system, to improve the overall health outcomes in the country.

This study has several strengths. This study provided evidence of the positive effect of the National Health Insurance (*Kor Por Sor*). Moreover, by using mixed methods where quantitative data and qualitative data were collected and interpreted together, this study provided a more comprehensive picture of the health-seeking behaviour situation in Lao PDR, compared with using quantitative data or qualitative data alone.

However, the results of this study must be interpreted considering several limitations. First, in the FGDs and IDIs, social desirability might have affected the responses of both the villagers and healthcare workers. However, to obtain an honest response and to increase the credibility of the data, the researchers have established a good working relationship with the villagers and healthcare workers. Second, the results of this study are not representative of all malaria endemic districts of Lao PDR with different ethnic groups and different geographical

terrains. However, transferability of the lessons learnt from this study is possible by focusing on the key common issues affecting other malaria endemic areas such as lack of health personnel and the need to improve the health-care facilities.

## CONCLUSIONS

Although the treatment seeking in public healthcare facilities in this study was high, the knowledge regarding malaria transmission was low. Moreover, both the villagers and healthcare providers expressed dissatisfaction with and the desire to improve the quality of the public health-care facilities in this study. The introduction of the National Health Insurance (*Kor Por Sor*) was regarded positively by both the villagers and the healthcare workers and this is a first step to achieving UHC in Lao PDR. Therefore, a more integrated approach to improve healthcare seeking and healthcare delivery focusing on the villagers, health-care providers and the facilities should be accelerated.

**Acknowledgements** We would like to thank Dr Chanthalangsouk Soukhavady, Dr Syda Keothavone, Dr Souphaphone Phimmasan and Ms Chansamone Khongkham for assisting in the data collection. Our gratitude also goes to the dedicated staff at the Center of Malaria Parasitology and Entomology, Savannakhet Provincial Health Office, and Thapangthong District Health Office for their support. We would also like to thank the villagers and healthcare workers who participated in this study.

**Contributors** KICO conceived and designed this study with the supervision from MJ. KICO, PK and SKe collected and analysed the data. MI, PB, SKa and MJ provided critical comments and support during data collection and analysis. KICO wrote the first draft of the manuscript. KICO is responsible for the overall content as guarantor. All authors read and approved the final version of the manuscript.

**Funding** This study was financially supported by the JICA/AMED SATREPS project for the Development of innovative research techniques in the genetic epidemiology of malaria and other parasitic diseases in the Lao PDR for containing their expanding endemicity. The funder had no role in designing the study, collecting, analysing and interpreting the data, preparing the manuscript, or decision to publish.

**Competing interests** None declared.

**Patient consent for publication** Not required.

**Ethics approval** This study was approved by the National Ethics Committee for Health Research (NECHR), National Institute of Public Health (NIOPH), Ministry of Health, Lao PDR (No. 094 NIOPH/NECHR) and the Research Ethics Committee of the Graduate School of Medicine, The University of Tokyo (No. 12038). Participation was voluntary and a written informed consent was obtained from all participants.

**Provenance and peer review** Not commissioned; externally peer reviewed.

**Data availability statement** Data are available upon reasonable request. Data are available upon reasonable request to the corresponding author.

**ORCID iDs**
Ken Ing Cherng Ong http://orcid.org/0000-0001-7196-7671
Masamine Jimba http://orcid.org/0000-0001-5659-3237

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
