## [Reviewer comments · BMJ Open]

ARTICLE DETAILS

TITLE (PROVISIONAL)	Health-seeking behaviors in a malaria endemic district in Lao People's Democratic Republic: a mixed methods study
AUTHORS	Ong, Ken; Khattignavong, Phonepadith; Keomalaphet, Sengdeuane; Iwagami, Moritoshi; Brey, Paul; Kano, Shigeyuki; Jimba, Masamine

VERSION 1 – REVIEW

REVIEWER	Abdul Manaf, Rosliza University Putra Malaysia, Community Health
REVIEW RETURNED	19-Jul-2021

GENERAL COMMENTS	Dear Authors, this manuscript reports an important study to understand the health seeking behavior in a malaria endemic district. I have raised several issues and comments for your clarification: 1. Please provide the results of the in-depth interviews with healthcare workers in the abstract.2. What was the sampling method used in the quantitative study? Please clarify whether there is any sample size calculation made prior to recruitment. What were the inclusion and exclusion criteria?3. Please elaborate the recruitment procedure for the healthcare workers who participated in the in-depth interviews. what were the inclusion and exclusion criteria?4. Please clarify how do you decide on the number of FGD and IDi to be conducted in this study?5. How do you ensure rigour in the qualitative study?6. For the quantitative results, please elaborate on the details of the multiple logistic regression analysis conducted in the study. What were the variables included in the model?7. The data integration need to be reviewed. The joint display in Figure 2 does not seem to describe similar point - how do you make a conclusion that people who went to the health clinics and the district hospital corroborates with the National Health Insurance? Similarly, please elaborate the comments related to Figure 3 of the joint display.8. Is there any information from the study that explains the health seeking behaviour related to symptoms and severity of illness?
--

REVIEWER	Yan, Shirley Johns Hopkins University Center for Communication Programs
REVIEW RETURNED	20-Sep-2021

GENERAL COMMENTS	Overall, an important paper to highlight people's understanding and health-seeking behavior using mixed methods.
--

	With regards to the abstract, I'd suggest to add in specific percentages from the quantitative components would be helpful. Other results can be briefly mentioned and acknowledged (e.g. perceptions of public hospital centers, transportation challenges, training of health staff). My main issue with this paper is that the results are not described in full. For the quantitative results, only a few sentences are used to describe the tables, when there are additional results that could be highlighted. Additional points can be mentioned succinctly. As for the qualitative results, again they are mentioned very briefly. Some of the quotes need additional explanation, and the explanations for the themes are quite little. An additional summary of what else was said by other groups for each theme would be helpful, rather than only highlighting what key quotes. A fuller description of the range of responses given for each theme would be helpful to show the breadth of perspectives and experiences. Finally, some of the descriptive statistics between the two villages vary quite a bit (e.g. gender of respondents, religion of participants) and a reflection of whether (and to what extent) this influences the results. For any other large differences in the data, it would be helpful for the authors to reflect on that. Attached are a pdf with additional comments throughout the document.
--	---

VERSION 1 – AUTHOR RESPONSE

Reviewer 1 (Dr. Rosliza Abdul Manaf, University Putra Malaysia)

	Reviewer Reports:	Author's response
1	Dear Authors, this manuscript reports an important study to understand the health seeking behavior in a malaria endemic district. I have raised several issues and comments for your clarification: 1. Please provide the results of the in-depth interviews with healthcare workers in the abstract.	Thank you very much for your review and feedback. They are very constructive and helpful to improve the manuscript. We have included a sentence to summarize the results of the in-depth interviews with the healthcare workers in the abstract in Lines 39-40: “Healthcare workers also mentioned the lack of manpower and equipment in the in-depth interviews.”
2	What was the sampling method used in the quantitative study? Please clarify whether there is any sample size calculation made prior to recruitment.	In this study, as we intended to include all the households in both villages, we did not do a prior sample size calculation. For recruitment, we informed the whole village of our study. In Village A, there were 215

	What were the inclusion and exclusion criteria?	households at the time of study and 202 representatives of each household participated. In Village B, 111 representatives out of 125 households participated. We included all those who agreed to participate and as this study was entirely voluntary, those who did not turn up were not in any way coerced to participate. We have clarified this in Lines 124-130: “Study participants in this study were villagers and healthcare workers aged 18 years and above living and working in the two villages. After Village A and Village B were chosen, the village headmen of each village were informed the purpose of this study. For quantitative data collection, the village headmen informed the whole village a few days before the study took place, and requested a representative of each household who agreed and were willing to participate in the study to come at the specified time and place (either at the village meeting room or the meeting room of the health center).”
3	Please elaborate the recruitment procedure for the healthcare workers who participated in the in-depth interviews. what were the inclusion and exclusion criteria?	Similar to the recruitment of the villagers, we intended to include all healthcare workers. In rural Lao PDR, usually a village health center is staffed by fewer than 5 healthcare workers, we intentionally included all. We have clarified this in Lines 133-135: “For the in-depth interviews (IDIs) of the healthcare workers, all the healthcare workers were approached and informed of the purpose of the study. All those who agreed to participate were interviewed.”
4	Please clarify how do you decide on the number of FGD and IDi to be conducted in this study?	In our study, the main criterion to decide the number of FGDs and IDIs was data saturation. After each FGD or IDI, we had a discussion about the content of the FGDs and IDIs and did preliminary data analysis to identify the main themes. Once we have reached a conclusion that data saturation has been reached, we stopped the FGDs. For IDIs, as we have decided to include all the healthcare workers in both villages, we interviewed all those who agreed. Similar to the FGDs of the villagers, we also concluded

		that data saturation has been reached for the IDIs. We have clarified this in Lines 169-172: “Three authors (K.I.C.O., P.K., S.Ke.) collected data until saturation was reached, meaning no new information emerging from the study participants even after additional FGDs or IDIs were conducted.”
5	How do you ensure rigour in the qualitative study?	To ensure rigor in the qualitative part, we followed the guidelines in the COREQ checklist (Attached as Supplementary 2). For example, all interviewers are experienced researcher who are well-versed in the local context of rural Lao PDR. In addition to that, we have also established trust among the villagers and healthcare workers through prolonged engagement in the village for a malaria project prior to this study. For the rigor in data analysis, we followed the steps outlined by Nowell et al. to ensure the trustworthiness of our analysis. For example, we had several discussions before finalizing the themes and this was an iterative process to obtain consensus among all authors who analyzed the data. We have clarified this in Lines 181-188: “Audio data were transcribed verbatim in Lao language by native Lao research assistants. All the transcripts were double checked for accuracy. Thematic analysis was used to analyze the data and was performed by three authors (K.I.C.O., P.K., S.Ke.). To ensure trustworthiness in the analysis, the steps outlined by Nowell et al.²³ were followed. NVivo 12 Pro (QRS International Pty. Ltd, Doncaster, Australia) software was used to expedite the data analysis process. Guidelines in the Consolidated Criteria for Reporting Qualitative Research (COREQ) were followed (Supplementary 2).²⁴”
6	For the quantitative results, please elaborate on the details of the multiple logistic regression analysis conducted in the study. What were the variables included in the model?	As per your suggestion, we have added a sentence in the methods in Lines 160-162: “In both models, the explanatory variables were village, sex, ethnicity, malaria

		experience, and knowing that malaria is caused by mosquito bites.”
7	The data integration need to be reviewed. The joint display in Figure 2 does not seem to describe similar point - how do you make a conclusion that people who went to the health clinics and the district hospital corroborates with the National Health Insurance? Similarly, please elaborate the comments related to Figure 3 of the joint display.	In Figure 2, the quantitative constructs indicated that the proportion of villagers who would first seek treatment at the health centers or district hospitals when a family member falls sick was very high. The same could be said for malaria. However, we agree with you that this percentage might not have been due to only the introduction of the National Health Insurance. Nevertheless, after triangulating and integrating the quantitative component with the qualitative component, and analyzing both sets of data again, we came to the conclusion that this could be attributed to the introduction of the National Health Insurance. We added a limitation to acknowledge this point in Lines 53-54: “Baseline health-seeking behaviors data were not available but data integration indicated the improvement in health-seeking behaviors in the malaria endemic district.”
8	Is there any information from the study that explains the health seeking behaviour related to symptoms and severity of illness?	In Table 5, we described the results based on clinical vignettes of 5 common diseases and majority of all villagers answered public health facilities as the first choice to seek treatment. In the qualitative part, we did not find any discussions regarding health-seeking behaviors related to specific symptoms and severity of illness.

Reviewer 2 (Dr. Shirley Yan, Johns Hopkins University Center for Communication Programs)

	Reviewer Reports:	Author's response
1	Overall, an important paper to highlight people's understanding and health-seeking behavior using mixed methods. With regards to the abstract, I'd suggest to add in specific percentages from the quantitative components would be helpful. Other results can be briefly mentioned	Thank you very much for your review and feedback. They are very constructive and helpful to improve the manuscript.

	and acknowledged (e.g. perceptions of public hospital centers, transportation challenges, training of health staff).	We have added more information in the result section of the abstract as suggested Lines 34-41: “For malaria, 96.0% and 98.2% of villagers from Villages A and B respectively would first seek treatment at public health facilities. Villagers who have not experienced malaria before were more likely to perceive that the consequences of malaria were serious compared to those who have experienced malaria before (AOR=1.69, 95% CI: 1.03 to 2.75). However, qualitative data showed that villagers faced problems such as lack of medicines and medical equipment. Healthcare workers also mentioned the lack of manpower and equipment in the in-depth interviews. Nevertheless, villagers still preferred to seek treatment at the health center as the National Health Insurance was introduced.”
2	My main issue with this paper is that the results are not described in full. For the quantitative results, only a few sentences are used to describe the tables, when there are additional results that could be highlighted. Additional points can be mentioned succinctly.	We have included additional descriptions of the results as suggested. Lines 210-211: “In both villages, the majority of the villagers were farmers (Village A: 170 (84.2%); Village B: 105 (94.6%).” Line 221: “In both villages, no villagers attributed witchcraft to be the cause of malaria.” Lines 225-226: “Most villagers in both villages did not know the most effective treatment for malaria (Village A: 185 (91.6%); Village B: 85 (76.6%).”
3	As for the qualitative results, again they are mentioned very briefly. Some of the quotes need additional explanation, and the explanations for the themes are quite little. An additional summary of what else was said by other groups for each theme would be helpful, rather than only highlighting what key quotes. A fuller description of the range of responses given for each theme would be helpful to	We have included additional explanations and included more themes to show the breadth of perspectives as suggested, in the qualitative results section in Lines 250-326.

	show the breadth of perspectives and experiences.	
4	Finally, some of the descriptive statistics between the two villages vary quite a bit (e.g. gender of respondents, religion of participants) and a reflection of whether (and to what extent) this influences the results. For any other large differences in the data, it would be helpful for the authors to reflect on that.	We have added a reflection point in the discussion as suggested in Lines 357-359: “Although the two villages vary in terms of ethnic groups and religions, the health-seeking behaviors and the barriers they faced in receiving optimal healthcare services were similar.”
	Attached are a pdf with additional comments throughout the document.	We addressed your annotated comments in the PDF below
Introduction:		
1	specific numbers to share the prevalence of malaria in the GMS region would be helpful	We have added a sentence and a reference as suggested in Lines 67-69: “For example, from October-December 2018, the region reported 25,974 malaria cases, equivalent to a 21% decrease compared to the same period in the previous year.”⁶
2	is there any literature that can be cited around environmental factors? these are all largely individualistic, without recognizing that for example facilities may be hard, or these groups may be marginalized	We have added a sentence and a reference as suggested in Lines 79-80: “Access to routine healthcare services is also limited for these ethnic minorities due to geographical barriers.”⁷
3	such as? specificity would be helpful here	We have revised the sentence to make it clearer as suggested in Lines 82-83: “Moreover, many ethnic minorities are animists with their own unique beliefs towards illness such as evil spirits could cause malaria.”^{11 12}
Methods:		
4	was there a sample size target for both qual and quant?	In this study, as we intended to include all the households in both villages, we did not do a prior sample size calculation. For recruitment, we informed the whole village of our study. In Village A, there were 215 households at the time of study and 202 representatives of each household participated. In Village B, 111 representatives of 125 households participated. We included all those who agreed to participate and as this study was

		entirely voluntary, those who did not turn up were not in any way coerced to participate. We have clarified this in Lines 124-130: “Study participants in this study were villagers and healthcare workers aged 18 years and above living and working in the two villages. After Village A and Village B were chosen, the village headmen of each village were informed the purpose of this study. For quantitative data collection, the village headmen informed the whole village a few days before the study took place, and requested a representative of each household who agreed and were willing to participate in the study to come at the specified time and place (either at the village meeting room or the meeting room of the health center).” In our study, the main criterion to decide the number of FGDs and IDIs was data saturation. After each FGD or IDI, we had discussion about the content of the FGDs and IDIs and did preliminary data analysis to identify the main themes. Once we have reached a conclusion that data saturation has been reached, we stopped the FGDs. For IDIs, as we have decided to include all the healthcare workers in both villages, we interviewed all those who agreed. Similar to the FGDs of the villagers, we also concluded that data saturation has been reached for the IDIs. We have clarified this in Lines 169-172: “Three authors (K.I.C.O., P.K., S.Ke.) collected data until saturation was reached, meaning no new information emerging from the study participants even after additional FGDs or IDIs were conducted.”
5	where were the surveys and interviews conducted? How was confidentiality ensured if at all?	All surveys and interviews were conducted in either the village meeting room or the health center meeting room and we made sure that the confidentiality of the villagers were strictly maintained. We have clarified this in the Ethical considerations section in Lines 199-202:

		“Personally identifiable information was not collected and the IDIs and FGDs were conducted in a secured place with low risk of being overheard by outsiders, such as the village meeting room and the meeting room of the health center.”
6	what was their educational background in?	They were either medical doctors or healthcare workers. We have added this information in Lines 146-148: “Before data collection, six research assistants, who were medical doctors or healthcare worker, were trained on the content of the questionnaire and ethical issues such as privacy protection of the participants.” We also acknowledged the research assistants with their permission in the acknowledgement section of this manuscript in Lines 442-447:
7	period comes within quotes	We have revised accordingly.
Results:		
8	additional details for each could be provided given that the tables highlight many other statistics	We have included additional descriptions of the results as suggested. Lines 210-211: “In both villages, the majority of the villagers were farmers (Village A: 170 (84.2%); Village B: 105 (94.6%).” Line 221: “In both villages, no villagers attributed witchcraft to be the cause of malaria.” Lines 225-226: “Most villagers in both villages did not know the most effective treatment for malaria (Village A: 185 (91.6%); Village B: 85 (76.6%).”
9	additional details here describe if anything else was controlled for,	As per your suggestion, we have added a sentence in the methods in Lines 160-162: “In both models, the explanatory variables were village, sex, ethnicity, malaria

		experience, and knowing that malaria is caused by mosquito bites.”
10	were facilitators gender segregated too?	For the FGDs and IDIs, 3 authors conducted the interviews. All the sessions we led by two female authors who were local Laotian medical doctors with a male author as the note taker and recorder. As the research team has been working in the village for a few years before this data collection, all the villagers were familiar with the research team. We have clarified this point in Lines 169-172: “FGDs were conducted among villagers and IDIs among healthcare workers. Three authors (K.I.C.O., P.K., S.Ke.) collected data until saturation was reached, meaning no new information emerging from the study participants even after additional FGDs or IDIs were conducted.”
11	additional descriptions and summaries for each theme are required. I assume there are additional quotes beyond what is mentioned here as quotes?	We have included additional explanations and included more themes to show the breadth of perspectives as suggested, in the qualitative results section in Lines 250-326.
12	anything else aside from the training of health staff and the cleanliness of centers?	We have included additional explanations and included more themes to show the breadth of perspectives as suggested, in the qualitative results section in Lines 250-326.
13	additional explanation is needed for this-- it's not clear what this means under difficulties in getting to health services	We have included additional explanations and included more themes to show the breadth of perspectives as suggested, in the qualitative results section in Lines 250-326.
14	in a sentence like this, the themes should ideally all be mentioned here	We have revised accordingly as advised in Lines 306-309: “The major themes were “Mentioning about the National Health Insurance (“Kor Por Sor”)” and “Expressing hope to improve the health center.” Supplementary 6 shows all other themes that emerged during the IDIs with the healthcare workers.”
15	this is in conflict with the qual findings, when people said transportation was a challenge-- how do the authors reconcile this?	This is not in conflict with the qualitative findings. In our study, the village health facilities usage was high. However, these facilities were not equipped to handle complicated medical emergencies. Complicated cases that require intensive

		care are referred to the provincial hospital which is 5- to 7-hour drive in the dry season and longer in the rainy season. We have revised the text to make it clearer for the readers as advised. Lines 365-367: “The high usage of village public healthcare providers in this study might be due to the ease in accessing these public healthcare providers as the health centers of both villages were situated in the villages.”²⁹ In addition, we also have clarified this point in the integrated findings in joint display 3 (Figure 3).
--	--	--

VERSION 2 – REVIEW

REVIEWER	Yan, Shirley Johns Hopkins University Center for Communication Programs
REVIEW RETURNED	13-Nov-2021
GENERAL COMMENTS	Thanks so much for writing this manuscript-- it's an important contribution to the field to understand health-seeking behaviors and potential opportunities.